



# Spatial Patterns and Characteristics of Flood Seasonality in Europe

Julia Hall[1] and Günter Blöschl[1]

[1]Institute of Hydraulic Engineering and Water Resources Management, Technische Universität Wien, Vienna, Austria

*Correspondence to*: Julia Hall (hall@hydro.tuwien.ac.at)

**Abstract.**

In Europe, floods are typically analysed within national boundaries and it is not well understood how the characteristics of
local floods fit into a continental perspective. To gain a better understanding at the continental-scale, this study analyses
seasonal flood characteristics across Europe for the period of 1960-2010.

The timing within the year of annual maximum discharges or water levels of 4105 stations from a European flood database is
analysed. A cluster analysis is performed to identify regions with different flood seasons. The clusters are determined using
the monthly relative frequencies of the annual maxima, and are further analysed to determine the temporal flood
characteristics of each region and the European-wide patterns of bimodal and unimodal flood seasonality distributions.

Below 60° latitude, the mean timing of floods of individual stations transitions from winter floods in the West to spring floods
in the East. Summer floods occurring in mountainous areas interrupt this West to East transition. Above 60° latitude, spring
floods are dominant, except for coastal areas in which autumn and winter floods are observed. The temporal concentration of
flood occurrences around a specific time of the year is highest in North-Eastern Europe, with most of the floods being
concentrated within 1-2 months. The cluster analysis suggests that six regions with geographically distinct flood seasonality
distributions exist. Most of the stations (~73%) with more than 30 years of data exhibit a unimodal flood seasonality
distribution (one or more consecutive months with high flood occurrence). Few stations (~3%), mainly located on the
foothills of mountainous areas, have a clear bimodal distribution. Overall, the geographical location of a station in Europe can
give an indication of its flood characteristics throughout the year and is more relevant than catchment area and outlet
elevation for the observed flood seasonality.



## 1 Introduction

Understanding the spatial and temporal characteristics of floods across Europe is important for improving our understanding of the flood generation mechanisms and for enabling better flood estimation and forecasts. River floods in Europe are caused by several processes. The most common naturally occurring river floods are driven by rainfall (including rain on snow) and

snowmelt (sometimes combined with ice jams) and are modulated by soil moisture (Hall et al., 2014). Hence, depending on the time of the year (i.e. season) in which a flood peak occurs one can infer the hydrological processes that are likely to generate floods. For example, flood peaks occurring in late winter or early spring, together with rising temperatures, are likely to be snowmelt induced. A better knowledge of the flood seasonality can therefore assist in the identification of homogeneous regions with a dominant flood season for regional flood frequency analysis, the analysis of mixed flood frequency

distributions, and in the identification and attribution observed changes in flood discharges.

Previous research on flood seasonality in Europe has been limited by two main constraints. First, the focus of most studies has been at national scales or smaller regions, which limited the analysis to a relatively small and local set of flood-generating processes. For example, Beurton and Thieken (2009) determined three homogeneous flood regions in Germany when analysing the annual maximum floods (AMF) of 481 gauging stations. Similarly, Cunderlik et al. (2004) found three main

flood seasonality types in Great Britain examining 268 sites. A few studies analysed the flood seasonality at larger scales, for example Mediero et al. (2015) using 102 streamflow records within Europe, but with limited spatial coverage, and Blöschl et al. (2017) focusing on changes in flood seasonality. Second, most of the previous studies on flood seasonality focused on the mean date of the AMF occurrence and/or the temporal concentration of the floods around their mean date (e.g. Parajka et al. (2009) or Jeneiová et al. (2016) for both Austria and Slovakia), while the detailed characteristics of monthly flood seasonality

distributions has rarely been studied in Europe. However, if unimodal, bimodal or skewed seasonality distributions exist, the mean date of the AMF can be misleading and mask important insights into the flood generating mechanisms (Ye et al., 2017). It is therefore important to report not only the mean date to characterise flood seasonality, but to describe also in detail the temporal flood seasonality characteristics.

This paper examines the spatial and temporal patterns of flood seasonality at a continental scale, using an extensive database

that covers all climatic regions in Europe. The focus of this paper is on the identification of regions with similar seasonal flood characteristics and on the description of the full temporal distribution of the flood events within the year.

First, the study area and the European discharge data set used in this study are presented, followed by the analysis methods. In the results section, the spatial characteristics of the mean flood seasonality are presented together with an analysis of the seasonal flood characteristics across Europe. Spatial patterns and clusters are identified based on the monthly distribution of

AMFs. The clusters are then analysed in detail for their monthly flood seasonality distributions and spatial characteristics and, followed by a discussion of the results.





## 2 Study Area and Data

The data analysed here is based on the database presented by Hall et al. (2015), and has been updated as in Blöschl et al. (2017) with some additional updates later. The original database used as a starting point in this study includes data from over 5565 stations from 38 data sources (see Supplement for details).

This study encompasses hydrometric stations located within 6.5° W - 60° E and 29.25° N - 69.25° N (Fig. 1) with catchment areas ranging from 10 km² to 100,000 km² (Fig. 2b). The data consists of the dates of annual maximum discharge or annual maximum water level (daily mean or instantaneous values). The maximum of each year is based on the calendar year (January to December) with a few exceptions, which are based on the respective countries' hydrological year (which can start in September, October, or November). Only the annual maxima are analysed here, as the long-term mean of the flood timing is

more meaningful if a single flood per year is considered and, due to restrictions in data access and licensing for some areas and/or countries.

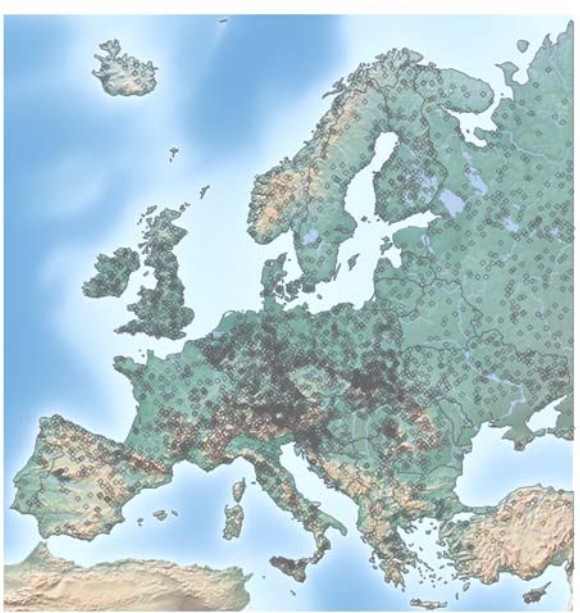

**Figure 1. Map of Study area, showing the topography and the location of the 4105 stations used in this study.**

Catchments for which it was evident that the flood timing is strongly affected by human modifications (e.g. dams or

15 reservoirs) are excluded from the analysis. All catchments with more than 10 years of data within the period 1960-2010 were included. In areas with high station densities, such as Austria, Germany, and Switzerland, only stations with at least 49 years of data in the analysis period were included to balance station density and to improve the visual representation on a European map. This selection resulted in 4105 hydrometric stations (Fig. 1) with record lengths ranging from 11 years to 51 years (Fig. 3), and station elevation ranging from -5.17 m to 1961 m (Fig. 2a).





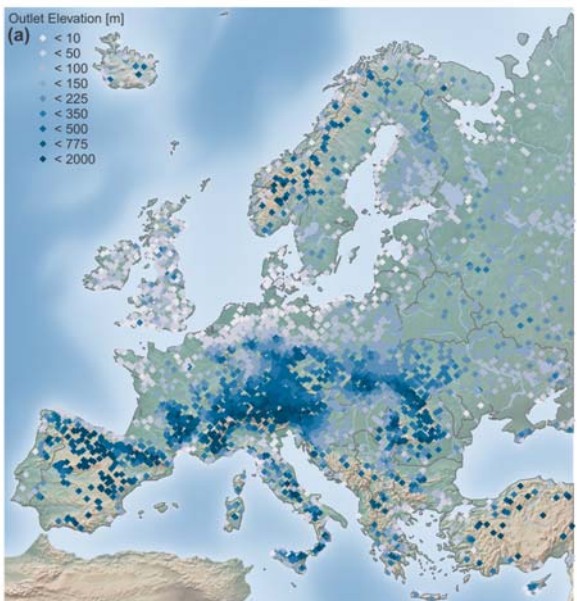
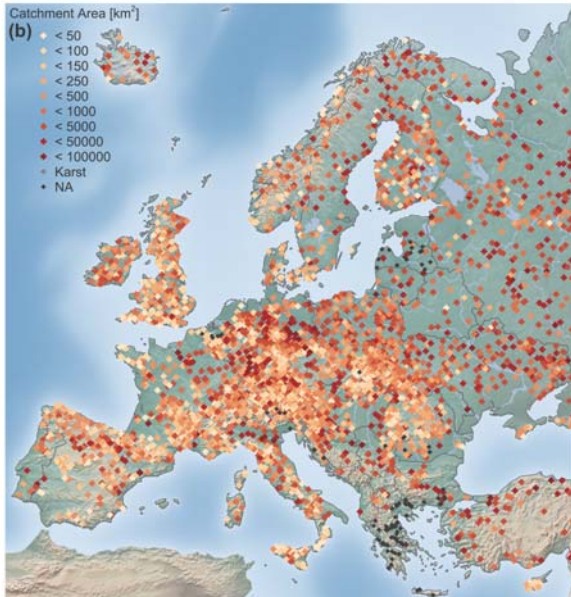

**Figure 2. Maps of station elevation at the catchment outlet [m] (a) and catchment area [km$^2$] (b).**

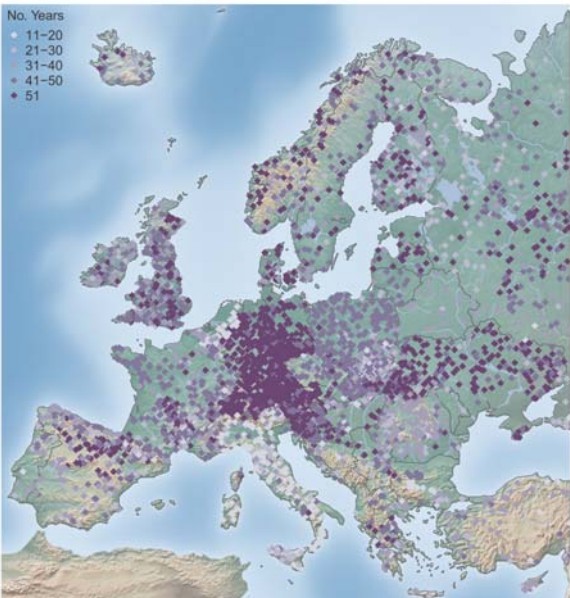

**Figure 3. Record length in number of years per station for the period 1960-2010.**





## 3 Methods

### 3.1 Flood seasonality

### 3.1.1 Mean flood seasonality and temporal flood concentration

The mean seasonality of annual maximum floods is determined using circular statistics (Bayliss and Jones, 1993; Mardia,

5  1972). In order to be able to calculate the mean date of flood occurrence $\overline{D}$ (i.e. day of year (DOY)) for a given station, the

date of the flood occurrence $D_i$ (DOY) in year $i$ is converted into an angular value $\theta_i$ in radians through

$$\theta_i = D_i \cdot \frac{2\pi}{m_i} \qquad 0 \leq \theta_i \leq 2\pi \tag{1}$$

where $D_i = 1$ corresponds to January 1 and $D_i = m_i$ for December 31, and where $m_i$ is the number of days in that year (365 or

366 for leap years). The mean date of occurrence $\overline{D}$ of a flood at a station is then

$$\overline{D} = \begin{cases} \left( \tan^{-1}(\frac{\overline{y}}{\overline{x}}) \right) \cdot \frac{\overline{m}}{2\pi} & \overline{x} > 0, \ \overline{y} \geq 0 \\ \left( \tan^{-1}(\frac{\overline{y}}{\overline{x}}) + \pi \right) \cdot \frac{\overline{m}}{2\pi} & \overline{x} \leq 0 \\ \left( \tan^{-1}(\frac{\overline{y}}{\overline{x}}) + 2\pi \right) \cdot \frac{\overline{m}}{2\pi} & \overline{x} > 0, \overline{y} < 0, \end{cases} \tag{2}$$

with

$$\overline{x} = \frac{1}{n} \sum_{i=1}^{n} \cos(\theta_i) \tag{3}$$

$$\overline{y} = \frac{1}{n} \sum_{i=1}^{n} \sin(\theta_i) \tag{4}$$

15  $\overline{x}$ and $\overline{y}$ are the cosine and sine components of the mean date, respectively, $\overline{m}$ is the mean number of days per year

(365.25), and $n$ is the total number of flood peaks at that station during the study period.

In order to be able to interpret the mean seasonality meaningfully, the Concentration Index $R$ of the dates of AMF occurrence

around the mean date is calculated.

20  $$R = \sqrt{\overline{x}^2 + \overline{y}^2} \quad 0 \leq R \leq 1 \tag{5}$$



R ranges from $R = 0$, representing no temporal concentration (i.e., floods are dispersed throughout the year and the seasonality vector of the individual floods cancel out (reflective symmetry)), to $R = 1$, which indicates that all floods occur on the same day of the year. R can be interpreted as a measure of how well the flood seasonality is defined (Fig. 4b).

There is a trade-off between retaining the best spatial coverage and the minimum record that is needed for meaningful flood seasonality analysis. Based on simulated monthly flood frequencies from a uniform distribution, Cunderlik et al. (2004) recommend care when evaluating the results from records shorter than 30 years, because of the large sampling variability.

In this observational dataset, the variability of the values of the flood Concentration Index R changes little with record length from 20 to 51 years (not shown). The R-values derived from records shorter than 20 years have a slightly higher variability than those of the longer records, but the difference is small. For the analyses of spatial patterns, priority is given to spatial coverage and all 4105 stations are used in the analysis of the mean seasonality, temporal flood concentration and the cluster analysis in. In the detailed analysis of the monthly flood characteristics (Section 3.3) only data with at least 30 years of record are used.

### 3.1.2 Circular uniformity

The spatial characteristics of a dominant flood seasonality can only be meaningfully interpreted if the data exhibit one or two preferred seasons in which floods occur (unimodal or bimodal flood seasonality). Therefore, stations for which the null hypothesis of circular uniformity (modified Kuiper's test (Mardia and Jupp, 2008) cannot be rejected ($\alpha$=0.1) (186 stations) are highlighted and analysed for their possible connection with spatial location (Fig. 6), catchment outlet elevation, and catchment area (Fig. 7). Only stations for which the null hypothesis of circular uniformity can be rejected are included in the remaining analyses (3933 stations), since the objective of the paper is the identification of clusters with distinct flood seasonality characteristics.

### 3.2 Cluster analysis

A cluster analysis is conducted to identify regions with similar flood seasonality across Europe. Depending on the method chosen, different regional clusters can emerge (Everitt et al., 2011). Here, the clusters are estimated using the k-means clustering algorithm. k-means can be considered superior to hierarchical clustering for our dataset, as k-means clustering is less affected by outliers and can be applied to large datasets, preferably for sample sizes > 500 (Everitt et al., 2011). More information on the k-means clustering algorithm by Hartigan and Wong (1979) used in the calculation (the function kmeans is part of the R package '*stats*') can be found in R-Core-Team (2016).

12 clustering variables are used, which contain the relative monthly frequency of flood occurrence for the months January to December. For each station, the monthly frequencies of the AMF are calculated. In order to reduce the influence of wide ranges between the variables of the k-means clustering, a Z-score standardisation of the variables is performed (Vesanto, 2001). Here, the monthly flood frequencies of all stations are standardised to zero mean and a standard deviation of one.





The standardised monthly flood occurrences are the only inputs to the k-means clustering algorithm. Geographic location is not used as a clustering variable to allow for an independent evaluation of the clusters, based on the time of flood occurrence only. Clusters consisting of stations with close geographical proximity or similar catchment characteristics may be considered more plausible than clusters for which this is not the case.

**3.2.1 Selection of the number of clusters**

One important step in clustering data is the decision on the number of clusters (*k*), as this number is not known *a priori*. In this study, different numbers of clusters are examined with the aim of obtaining homogenous groups (clusters) of stations that are as similar as possible (regarding the timing of flood occurrence) within their group but are also as dissimilar as possible from the stations not belonging to their group.

The performance of the k-means clustering algorithm is assessed using the silhouette value s(*i*) (Rousseeuw, 1987), which is a measure of how similar a station is to its own cluster compared to the other clusters. Silhouette values range from -1 (high similarity with the neighbouring cluster) to 1, with higher s(*i*) values indicating that the station has a high similarity to its own cluster.

For a number of *k* clusters (*k*>1) the silhouette value s(*i*) can be calculated using Eq. 6,

$$s(i) = \frac{b(i) - a(i)}{\max\{a(i), b(i)\}} \qquad (6)$$

where a(*i*) is the average dissimilarity of all variables (here the average Euclidean distance is used) of station *i* to all other stations in the same cluster (i.e. how distant the station is, on average, from the other stations) and b(*i*) is the average dissimilarity to all stations in the neighbouring cluster to station *i* (i.e. the cluster that has the lowest average dissimilarity

from all other clusters). The average silhouette value over a cluster ($\bar{s}(i)_{cluster}$) thus indicates how similar the stations in a cluster are, and the average silhouette value $\bar{s}(i)$ over all stations in the dataset indicates how well the clustering algorithm has assigned the stations to their respective cluster. The number of *k* clusters that has both the highest $\bar{s}(i)$ and highest individual $\bar{s}(i)_{cluster}$ can be considered the best choice (Rousseeuw, 1987).

As a second criterion for the selection of *k* clusters, the 'Elbow method' based on the total sum of within-cluster sum of squares (TSS$_{within}$) is used (Equation 7),

$$TSS_{within} = \sum_{j=1}^{k} \sum_{i=1}^{n_j} (Y_{ij} - \bar{Y}_j)^2 \, , \qquad (7)$$

where *k* is the number of clusters, *j* is a specific cluster, and *i* is a individual station in that cluster, so that $Y_{ij}$ is the *i*[th] observation in cluster *j*. $\bar{Y}_j$ is the average of $Y_{ij}$ over the range of *i*.





With an increasing numbers of clusters $k$, the TSS$_{within}$ decreases. The optimal number of clusters is determined using the magnitude of the reductions in the TSS$_{within}$ between two consecutive clusters. If the reductions do not decrease much beyond a certain number of clusters, that number is considered a good choice. After accounting for the sensitivity of the initial centroid placements (see below), the final number of clusters is selected based on first the $\bar{s}(i)$ values and second the Elbow

method conditional on the TSS$_{within,}$ values.

The k-means clustering algorithm is sensitive to the location of the initial $k$ centroids to which the nearest neighbours are assigned (Steinley, 2003). This sensitivity affects both the selection of the 'optimal number' of clusters k and the cluster assignment itself. To account for this, the k-mean algorithm is repeated with 10,000 random centroids initialisations (seed vectors) and the initialisation with the highest average silhouette value over all stations $\bar{s}(i)$ is selected. As several initial

centroid locations for $k$ clusters can result in the same maximum $\bar{s}(i)$ values, all centroids initialisations that have the same $\bar{s}(i)$ values are retained and further analysed with regard to their TSS$_{within}$ values. From these initialisations, only the sets of initial centroids that have the same optimal number of clusters $k$ based on the $\bar{s}(i)$ values and the evaluation of the TSS$_{within}$ values are retained as described above. As this can result into more than one set of initial centroids, the set that has the lowest TSS$_{within}$ of the remaining sets is chosen as the final location of the initial centroids.

**3.3. Analysis of monthly flood characteristics**

**3.3.1. Identification of flood dominant and flood scarce months**

The $k$ clusters obtained are then further analysed for their temporal characteristics, with the aim of identifying months of high flood occurrence and months in which floods occur seldom or never (hereafter termed flood dominant and flood scarce months, respectively). This classification into flood dominant and flood scarce months is achieved by a significance test, in

which the observed monthly flood occurrence is compared to the expected occurrence of a uniform flood seasonality distribution (1/12 of the floods are expected to occur in each month), (Cunderlik et al., 2004).

As the months contain a different number of days, the monthly counts of flood occurrence $c_i$ need to be modified to '30-day months' to obtain adjusted monthly percentages of flood occurrences ( $\widetilde{l}_i$ ). $\widetilde{c}_i$ is the adjusted monthly count of flood occurrences with $i$ being the months 1 to 12, and $d_i$ the number of days in that month (February has 28.25 days to account for

leap years).

$$\widetilde{c}_i = c_i \cdot \frac{30}{d_i} \tag{8}$$

$$\widetilde{n} = \sum_{i=1}^{12} \widetilde{c}_i \tag{9}$$

$$\widetilde{l}_i = \frac{\widetilde{c}_i}{\widetilde{n}} \cdot 100 \tag{10}$$





The one-sided 95% upper ($L_{upper}^n$) and lower ($L_{lower}^n$) confidence intervals are approximated following Cunderlik et al. (2004):

$$L_{upper}^n = \frac{n + 11.491}{0.048\, n^{-1.131}} \tag{11}$$

$$L_{lower}^n = \frac{n - 27.832}{0.199\, n^{-0.964}} \tag{12}$$

with $n$ being here the record length.

If the monthly percentage $\widetilde{l_i}$ of a given month is above or below the confidence interval, this month is considered to be either flood dominant or flood scare respectively (at a 5% significance level). Only stations with least 30 years of data are analysed (3356), as the above approximation is only valid for records with at least 30 data points. The 563 stations with shorter records

are excluded from the remaining analyses.

Depending on the record length, the upper and lower thresholds of the confidence interval vary. For example, for a 30-year long record, the $L_{upper}^n$ and $L_{lower}^n$ for $\widetilde{l_i}$ would be 0.246% and 10.126% respectively (i.e. $\widetilde{c}$ counts of flood occurrences for a given month of 0.073 and 3.037) whereas for a 51-year long record the thresholds for $\widetilde{l_i}$ would be 2.629% and 15.251% respectively (i.e. $\widetilde{c}$ counts for a given month of 1.341 and 7.778). The months that have their $\widetilde{l_i}$ within these thresholds are

not further classified here. For each station independently, each month of the year is classified as flood dominant, flood scarce or neither of them (i.e. unclassified).

### 3.3.2. Identification of bimodal and unimodal flood seasonality distributions

Flood dominant or flood scarce periods for a station are obtained by segmenting the year based on the consecutive occurrence

of months with the same classification (i.e. either flood dominant or flood scarce). If the months at the beginning and the end of the year belong to the same classification, the months are combined to form one consecutive period. The length of the periods is determined by summing the number of months within each individual period.

Based on these periods, the monthly flood seasonality distribution is identified as bimodal if two flood dominant periods, independent of their length (i.e. a minimum of one month each), are separated by at least one flood scarce month (before and

after). A unimodal flood seasonality distribution is identified if all months considered as flood dominated occur consecutively.



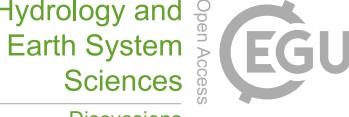
## 4 Results

### 4.1 Flood Seasonality Analysis

Figure 4 shows the mean flood seasonality and the temporal concentration of flood occurrence within the year. A distinct spatial pattern of the mean timing of floods within the year can be observed (Fig. 4a). Below 60° latitude, the mean

seasonality transitions from winter floods in the West to spring floods in the East due to increasing continentality. Stations located in mountainous areas (e.g. the Alps, the Carpathians, and the Pyrenees) exhibit predominately summer floods and disrupt the West to East transition of the flood timing. Above 60° latitude, spring floods dominate the spatial pattern, except for coastal areas in which autumn and winter floods are observed. The temporal concentration of floods around the mean date of flood occurrence (R-value) (Fig. 4b) is highest in North-Eastern Europe. High temporal concentration is also apparent at

the western coast of Europe except for the northern coasts where floods are spread more evenly throughout the year. Catchments on the foothills of mountainous areas (e.g. around the Alps and the Carpathians) also tend to have smaller R-values. The orange crosses in Fig. 4b indicate the stations for which circular uniformity could not be rejected at a significance level of α=0.01. The characteristics of stations with uniform flood occurrence throughout the year are later examined in detail (e.g. Fig. 6).

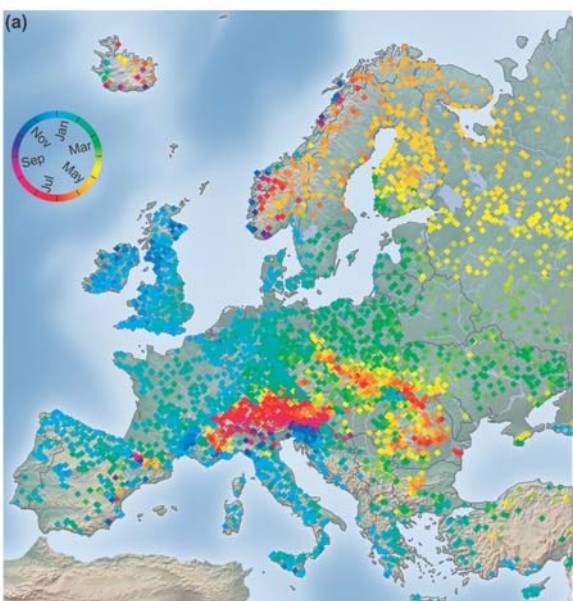
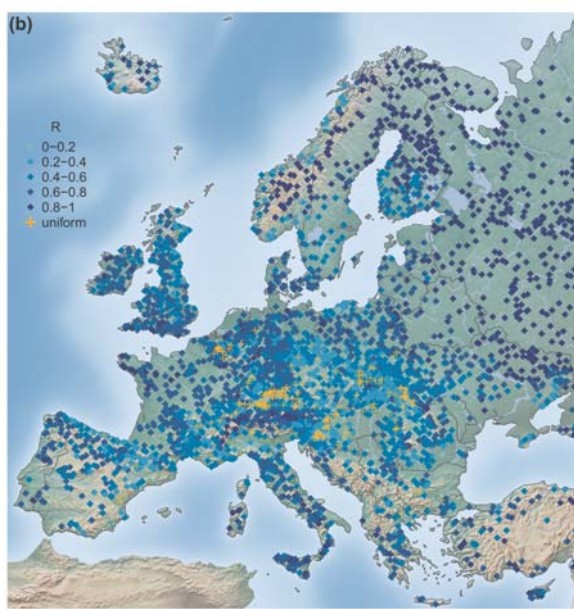

**Figure 4. Seasonality of floods in Europe for 1960-2010. Mean date of flood occurrence $\overline{D}$ (a). Flood Concentration Index R (b); Stations with circular uniformity are marked by orange crosses.**



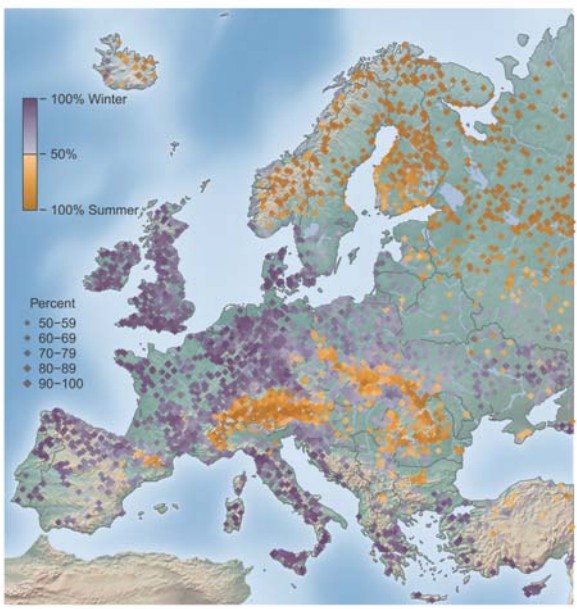

**Figure 5. Percentage of winter half-year (October to March) and summer half-year (April to September) floods. Dark purple/orange colours indicate dominance of the winter/summer half year, light colours indicate an almost equal occurrence in the two half-years.**

An alternative way of examining the flood seasonality is the frequency of floods occurring in the winter and summer half years (Fig. 5). The winter and the summer half-years are defined as October-March and April-September, respectively. There is a clear dominance of summer floods in the North and North-Eastern parts of Europe, which can be characterised by a continental climate and in the mountain ranges (Pyrenees, Alps, Carpathians). In the rest of Europe, floods predominately occur in the winter half-year. Transitional areas for which no clear seasonal distinction can be made (< 60 % of either winter

of summer half-year floods) can be found in and around Poland, Lithuania, Belarus and parts of the Ukraine. In these transitional areas, no half-year flood season dominates, as the AMF of these stations tend to occur in March and April around the cut off date separating the winter- versus summer half-years. Additionally, a less clear flood seasonality can be found on the foothills of mountains, where both winter and summer floods occur (mixed distribution), depending on whether floods are snowmelt induced, summer rainfall induced, or the floods are uniformly distributed around the year.

In order to further examine the relationship between week seasonality (low R-values) and uniform flood occurrence, the location of the stations for which circular uniformity could not be rejected at a significance level of α=0.01 is shown in Fig 4b. The stations with a uniform flood seasonality distribution are found predominately in small catchments and at low to medium high altitudes (< 1000 m) (Fig. 6a). However, for some of the stations with a small Flood Concentration Index R, uniformity could not be rejected for the significance level α=0.01, which reveals that small R-values do not necessarily

indicate uniformity. These stations likely possess a skewed or a bimodal distribution of flood occurrence throughout the year.





For the European continent, stations with high station elevations tend to have a high Flood Concentration Index R and occur mainly in early summer (mean seasonality in May and June) (Fig 6a). At higher elevations, there are no stations with uniform flood occurrence, whereas at lower elevations (< 1000m) uniform distributions do exist. Uniformity is not rejected for catchments of all sizes. Large catchments indicate uniformity, but catchments with less than 1000 km$^2$ exhibit more often smaller R-values and these tend to have uniform distributions (Fig. 6b). Overall, uniformity of flood occurrence seems to be predominately conditioned by geographical location (foothills of mountains) (Fig. 2b), which is related to catchment elevation if these stations are near mountain ranges.

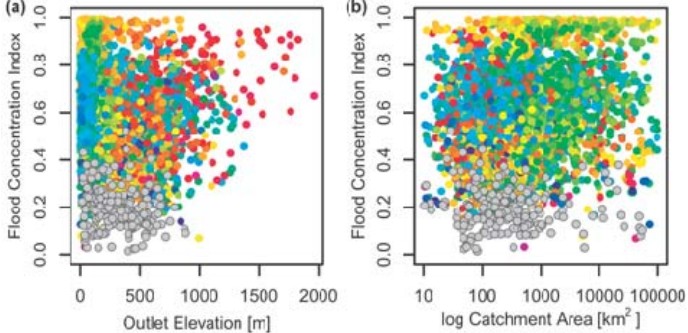

**Figure 6. Flood Concentration Index R of floods in Europe (1960-2010) dependent on station elevation (a) and catchment area (b). Colour of points indicates the mean timing of floods at that hydrometric station location (as in Fig 4a). Grey points indicate the stations for which circular uniformity could not be rejected.**

The mean frequency of floods in each season (based on individual flood events) is shown in Fig. 7. Floods occurring between January and March are classified as winter floods, spring floods occur between April and June, summer floods between July and September, and autumn floods between October and December. Figure 7a, displays an increase in the mean frequency of summer floods with increasing elevation and conversely a tendency towards decreases in the frequency of autumn and winter floods due to the increasing dominance of summer floods (see also Fig 6a). Autumn floods have the highest frequency in most of the elevation ranges. In two elevations ranges (91 to 125 m and > 440 m), spring floods have the highest occurrence frequency. Figures 4a and 5, suggest that the high mean frequency of spring floods either occurs in catchments with intermediate elevation in North-Eastern Europe or in, or around, mountainous areas (the timing is often towards the end of June, close to July which is the first month used for the classification of summer floods). Overall, smaller catchments in Europe are more similar regarding their mean frequency of seasonal floods (Fig. 7b). With increasing catchment area, the percentage of spring floods increases. This observed tendency is related to the uneven spatial distribution of larger catchments in the database (Fig. 2b). Stations with large catchments areas can be found predominately in the central and eastern to north-eastern parts of the study area, which are dominated by winter and spring floods.





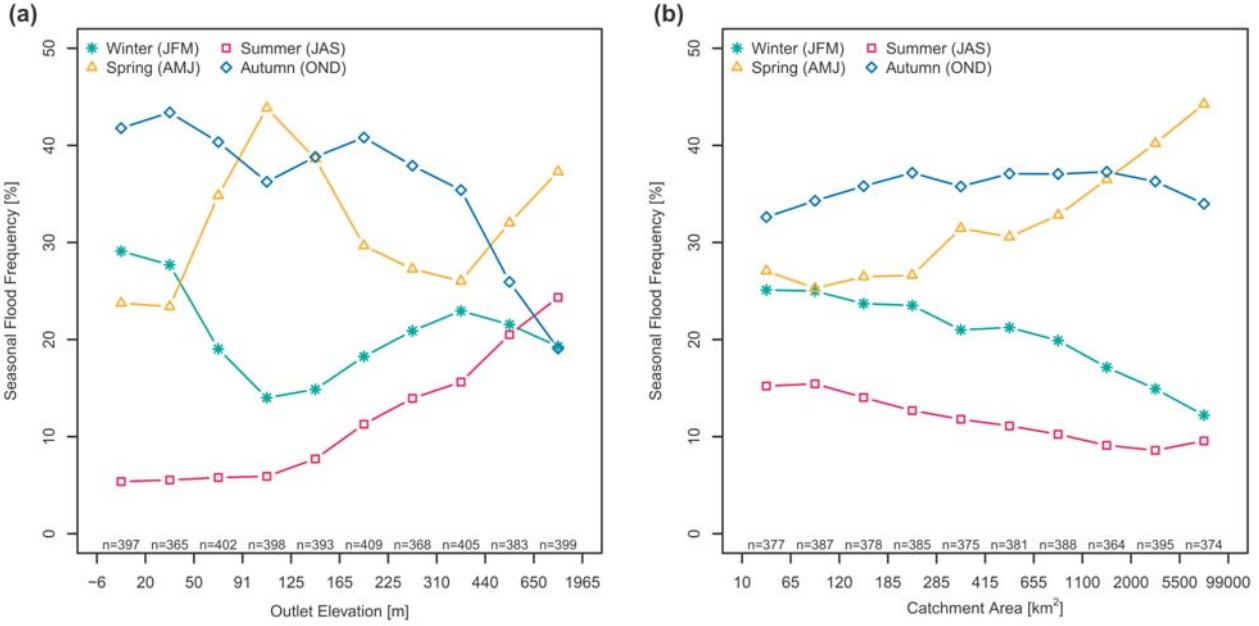

**Figure 7. Mean frequency of seasonal floods by ranges of outlet elevation (a) and catchment area (b). In both panels, the ranges on the x-axis were selected so that roughly an equal number of stations is allocated to each range.**

## 4.2 Cluster Analysis on Flood Seasons

In the previous section, the strong influence of the geographical location on the timing of flood occurrence at a given station is apparent. Therefore, it is of interest to identify larger scale regions with relatively similar seasonal flood occurrence in Europe. These regions are identified with the help of cluster analysis after the best possible initial centroid locations are determined. Table 1 summarises the sensitivity of the location of the initial centroids and shows the percentage of how often a specific number of clusters ($5 \leq k \leq 7$) obtained the highest average silhouette value $\bar{s}(i)$ from the 10,000 random initial cluster centroids and the highest overall $\bar{s}(i)$ value. Table 1 indicates that, with the same initial centroid placement for 5, 6 or 7 clusters (same as 5 clusters plus one or two additional initial centroids for 6 and 7 clusters respectively), 46% of the random samples generated the highest $\bar{s}(i)$ values for 6 clusters. Additionally, the 6 initial cluster centroids resulted in a clustering that obtained the maximum $\bar{s}(i)$ of 0.442 for all 10,000 random initialisations. In the initialisations for which 5 or 7 clusters obtained the highest $\bar{s}(i)$, the $\bar{s}(i)$ were always lower than the one obtained with 6 clusters. Therefore, the sets of initial centroid locations that obtained the highest $\bar{s}(i)$ of all random initialisations (0.443) for $k$=6 were chosen as the candidates for further selection of the initial centroid position. From these only the sets of initial locations are retained for which the Elbow method (based the reduction in the total within cluster sum of squares (TSS$_{within}$)) also resulted in 6 optimal clusters. As several sets with different initial centroid locations fulfilled this criterion, the initial set of centroids that yielded the lowest TSS$_{within}$ for $k$=6 is selected and is used in the remainder of the study.





**Table 1. Number of clusters and average silhouette value $\bar{s}(i)$.**

| Number of clusters ($k$) | Samples with highest $\bar{s}(i)$ | Maximum average value $\bar{s}(i)$ |
|---|---|---|
| 5 | 39 % | 0.438 |
| 6 | 46 % | 0.443 |
| 7 | 15 % | 0.396 |

Figure 8) depicts the spatial distribution of the six clusters of monthly flood occurrences. Most clusters are spatially coherent except Cluster 4. Table 2 and Fig. 9 assist in interpreting these clusters. Cluster 1 is located in Western, Central and Southern Europe and contains most of the stations (~36%). The mountainous regions in Europe (highest average outlet elevation), the Alps and the Carpathian and Scandinavian Mountains, in Cluster 2 account for ~15% of the stations. Most stations located in Central and Eastern Europe up to 55°N (~24%) are assigned to Cluster 3. Cluster 5 and 6, located predominately in Northern and North-Eastern Europe, are the two smallest clusters containing ~8% and ~6% of all stations respectively. Most of stations assigned to Cluster 6 are located above 60°N and are low lying.

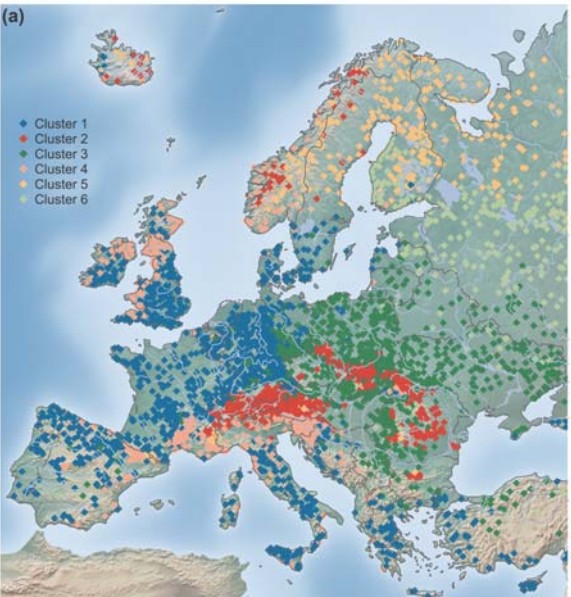
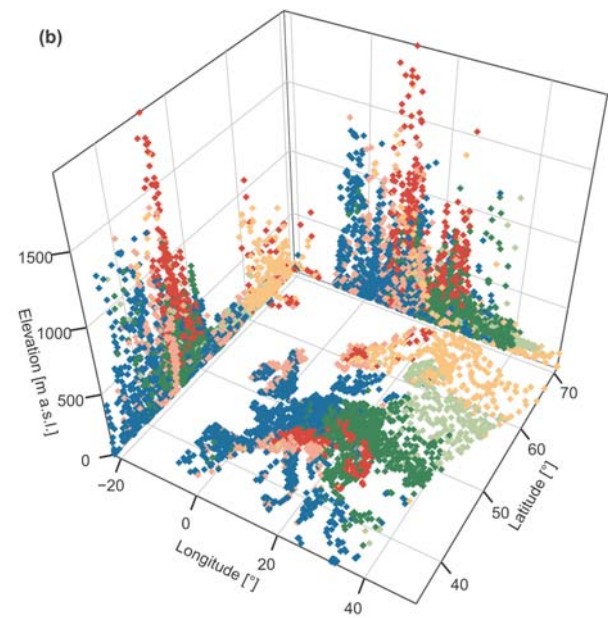

**Figure 8. Spatial distribution of the six clusters of monthly flood frequencies (a). The vertical axes of the panel on the right shows the catchment outlet elevation (b).**



**Table 2. The six clusters of monthly flood frequencies in Europe and their characteristics.**

| Number of cluster ($k$) | Location | Number of stations | Average silhouette width $\bar{s}(i)$ | Average station elevation [m a.s.l.] | Average catchment area [km²] | Average of all $D_i$ (DOY) | R-value over all $D_i$ |
|---|---|---|---|---|---|---|---|
| 1 | Western, Central and Southern Europe | 1427 | 0.51 | 220.9 | 2193.0 | 25 Jan (25) | 0.60 |
| 2 | Mountainous regions | 595 | 0.40 | 538.8 | 2010.1 | 30 June (181) | 0.53 |
| 3 | Central and Eastern Europe | 934 | 0.37 | 207.6 | 3950.8 | 22 Mar (81) | 0.51 |
| 4 | Western British-Irish Isles, Western Coast of Norway and Northern Mediterranean | 405 | 0.26 | 263.8 | 1757.9 | 5 December (339) | 0.36 |
| 5 | Northern Europe | 307 | 0.56 | 204.9 | 3940.4 | 19 May (139) | 0.85 |
| 6 | North-Eastern Europe | 251 | 0.62 | 126.2 | 6607.4 | 15 April (105) | 0.84 |

Cluster 1, 5 and 6 are well defined (i.e. high within-cluster similarity or average silhouette width $\bar{s}(i)$ ). Cluster 4 is the least well-defined cluster in terms of $\bar{s}(i)$ and also in terms of spatial location. The stations in Cluster 4 are found in several regions of Europe (Western British-Irish Isles, Western Coast of Norway and Northern Mediterranean), and have the smallest catchment average size and the largest variety of mean seasonality of individual stations. The largest catchment areas are found in Northern and North-Eastern Europe (Cluster 3, 5 and 6). The average seasonality characteristics ($\bar{D}$ and R-value) of the clusters, based on all flood dates ($D_i$) (see also 'mean of all' in Fig 10), identify Cluster 4 as the cluster with the highest spread of flood occurrence around the mean date of flood occurrence (early December).

The average dates of the flood timing in Clusters 5 and Cluster 6 are mid-May and mid-April, respectively, with all floods being highly concentrated around the average date. Cluster 1 is also strongly seasonal with a mean flood occurrence in late January, whereas the mountainous areas (Cluster 2) have their mean flood occurrence in summer. Overall, the geographical location seems to determine the membership of a cluster, although the geographic location is not included as a variable for clustering. There are a few stations that do not fit the large-scale, coherent cluster pattern (i.e. spatial outliers).





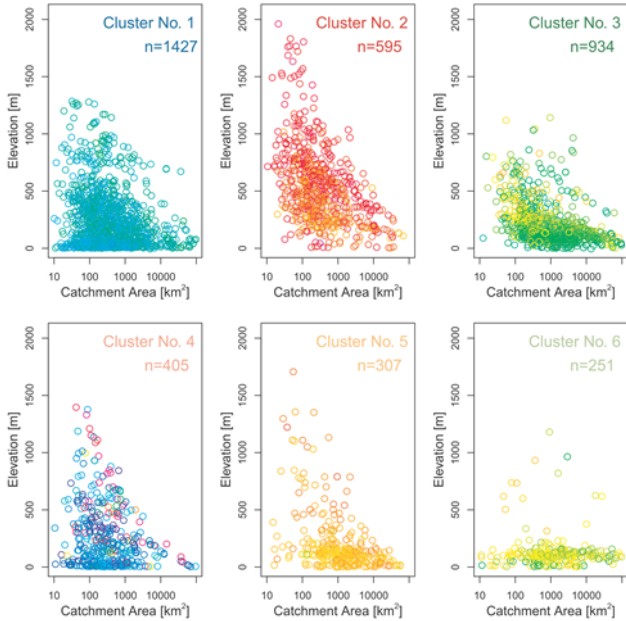

**Figure 9. Mean flood seasonality as a function of catchment outlet elevation [m a.s.l.] and catchment area [km²] grouped by the six clusters. Colours of the points correspond to the mean seasonality of Figure 4a.**

Fig. 10 shows the mean flood seasonality for each station, the overall mean seasonality of all floods belonging to the same

cluster, the mean of all mean flood seasonalities, as well as the respective temporal concentration around these means. The stations within their respective clusters display similar concentrations, as indicated by R-values > 0.9 of the mean of the cluster mean flood seasonalities. The exception is Cluster 4, which has the lowest temporal concentration of the mean floods with R=0.71. In this cluster, the temporal concentrations of floods of the individual stations are lower than those of the stations in the other clusters. The R-values of the mean of all AMF in both Clusters 5 and Cluster 6 (R=0.85 and R=0.84,

respectively) are close to the mean of all mean seasonalities. This indicates that, in these clusters, not only the mean seasonalities are temporally concentrated but also the individual floods. The mean seasonalities of most of the stations assigned to these clusters have a strong temporal concentration around their regional mean (high R-value), and only a few stations have a larger spread around the mean date of flood occurrence. The R-values of the regional mean seasonality of all AMF in the other clusters are much smaller than the R-values of the mean of all mean seasonalities. This indicates that the

clustering algorithm performs well with regard to clustering stations that have a similar mean seasonality, but individual flood events may exhibit higher temporal variability.





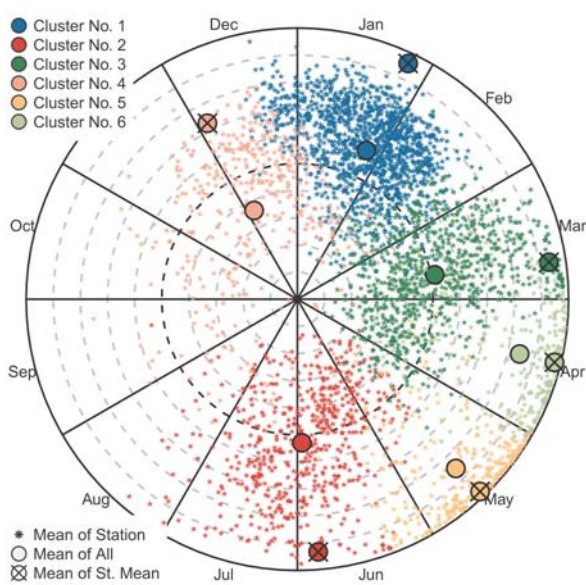

**Figure 10. Mean Seasonality and temporal concentration of floods for each station (small points), the mean over all floods within specific clusters (large points) and the mean of all mean flood seasonalities (large points with crosses) within specific clusters. Colours correspond to clusters. Distance to centre is a measure of the temporal Flood Concentration Index R, with the centre corresponding to R=0, the black dashed circle to R=0.5 and the outer full circle to R=1. The grey dashed circles correspond to intervals of 0.1 R.**

The mean seasonality has limited information content, as it only reflects the first moment of the seasonality distribution. Therefore, it is of interest to examine the full monthly distribution. Figure 11 shows the relative monthly frequency of AMF of the cluster centroids (CC) (i.e. the cluster means). If all floods were equally spread over the year, one would expect all months to contain about 8.3% of all the AMF. All CC have at least 1 month, in which more than 18 % of the annual maximum floods occur, which indicates that there is a clear dominant month for floods to occur and therefore clear flood distinct flood seasonality in each of the CC. CC 5 and CC 6 have the months with the highest frequency of flood occurrences in a single month with >60% of the AMF (May and April, respectively); the previous and following month account for ~10% and 15%, respectively. The rest of months have less than 3 % of the AMF. CC 1 and CC 2 exhibit an almost bell-shaped distribution with the AMFs peaking in the winter and summer half of the year, respectively. CC 3 peaks at the beginning of the year (strongest peak in May (28 %)) with few AMFs in the rest of the year. CC 4 is the cluster with the least pronounced peak in the monthly flood frequencies. There is a small peak in the winter (October to September) and a low flood season in the summer months. September is the month, in which all CC exhibit the lowest number of floods (Fig. 11).



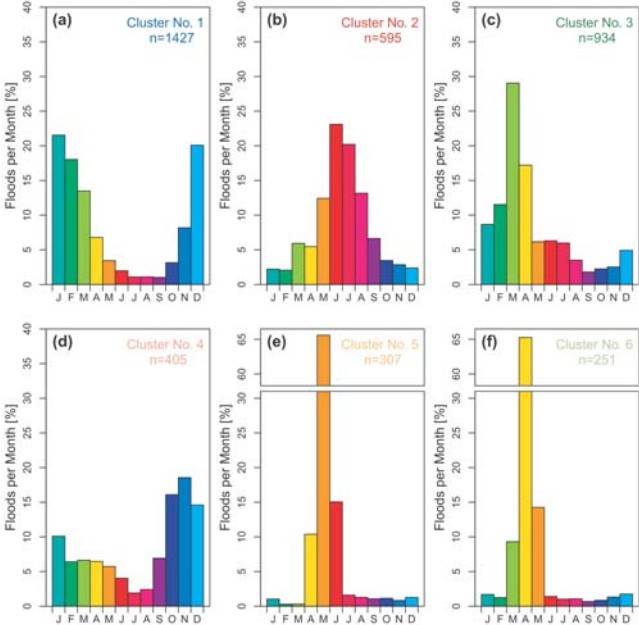

**Figure 11. Frequency of occurrence of maximum annual floods by month for the cluster centroids CC (i.e. cluster means).**

Fig. 12 depicts the full range of relative monthly flood frequencies of all stations assigned to each cluster. The shape of the circular distribution of the medians of each cluster resembles the shape of the cluster centroids in Fig. 11. However, for some

clusters, the monthly flood seasonality distributions differ from the CC in Fig. 12. This is of particular importance for stations that can be considered outliers.

In Cluster 1, the shape of the distribution remains similar, however for individual stations (outliers) the winter months have a much higher percentage of flood occurrences (up to 55%). The summer months stay below 15% of the AMF in Cluster 2, and for most of the stations June and July remain the months with the highest percentages. For some stations, August and to a

lesser extent May, are also important. This characteristic could not have been detected when examining the CC in Fig. 11 alone. October to February have low flood occurrences in Cluster 2.

Within Cluster 3, March and April stand out as the most important months of flooding. These months also have the highest spread between stations, while the other months have similar spreads. In Cluster 4, the CC show frequent floods in October to January. However, when taking into account the characteristics of all the stations, the months April, May, and June also

contain a high percentage of AMF (up to 55 %), depending on the station. This characteristic is not expressed in neither the mean nor the median of the cluster (Fig. 11d and 12d) respectively) and indicates that, for some stations (i.e. locally), these months are important in terms of flooding. The appearance of months with an additional secondary peak in flood occurrence indicates the possible existence of a bimodal distribution for several stations.

The medians and all stations of Cluster 5 and Cluster 6 exhibit a high occurrence of flooding in May and April, respectively.

Most other months of the year show very low frequencies for the median with a very low spread between stations. The





exceptions are the months immediately before and after the main flood month, which can have high percentages of flooding for individual stations as well. This high concentration of floods around a single month is the reason for the stations, belonging to these clusters, to show very high R-values in Fig 4b. Overall, when taking all stations within a cluster into account, one can see that the characteristics that are present in the monthly distributions of the CC are retained. However, 5 some additional characteristics such as the emergence of a bimodal distribution in Cluster 4, which was smoothed-out in the CC or additional months with higher relative monthly flood frequencies and outliers, can be identified.

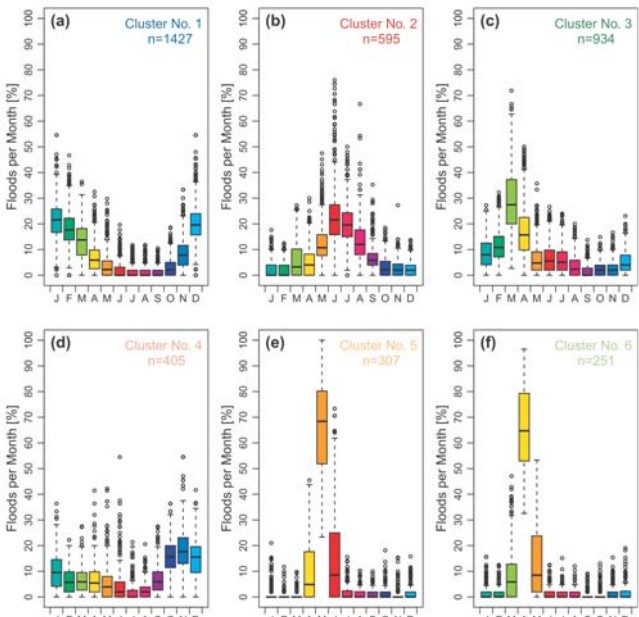

**Figure 12. Boxplot of the percentage of floods per month for each station, grouped by cluster (a-f). The top and bottom of the boxes show the 75th and 25th percentiles (i.e. the upper and lower quartiles) respectively, whiskers extend to 1.5 × interquartile range** 10 **beyond the box, the black band indicates the median, and outliers are shown as points.**

In a next step, the classification into flood dominant and flood scare months is performed, on records with more than 30 years of data ($n_{sub}$). Each cluster has more than 80% of their stations with series longer than 30 years (see Table 3 for the exact numbers). Figure 13 shows the percentages of stations in each cluster for which a specific month can be considered 15 statistically significant ($\alpha=0.05$) as being flood dominant or scarce. In Cluster 5 and Cluster 6, the months May and April respectively are flood dominant for all stations (100%). There are four clusters with at least one month that can be considered not to be flood dominated in any station (marked by stars above the x-axis in Fig 13). In Cluster 1, this is the month June and September, in Cluster 3, September, in Cluster 5, February and March, and August to October and in Cluster 6, August to November. In Figure 13 there is not a single month, in any of the clusters, for which all stations would exhibit flood scarcity 20 (i.e. 100%). All clusters (apart from Cluster 2) have at least one month in which no station has scarce flood occurrence



(marked by stars below the x-axis in Fig 13). This is February for Cluster 1, March for Cluster 3, October and November for Cluster 4 and March and April for Cluster 5 and Cluster 6 respectively. Based on the percent of stations that have flood dominance in Cluster 4 there is again an indication that some of the stations might have a bimodal distribution with a secondary peak around April and May (12.01% and 10.21% of the stations have a significant flood dominated months

5    respectively).

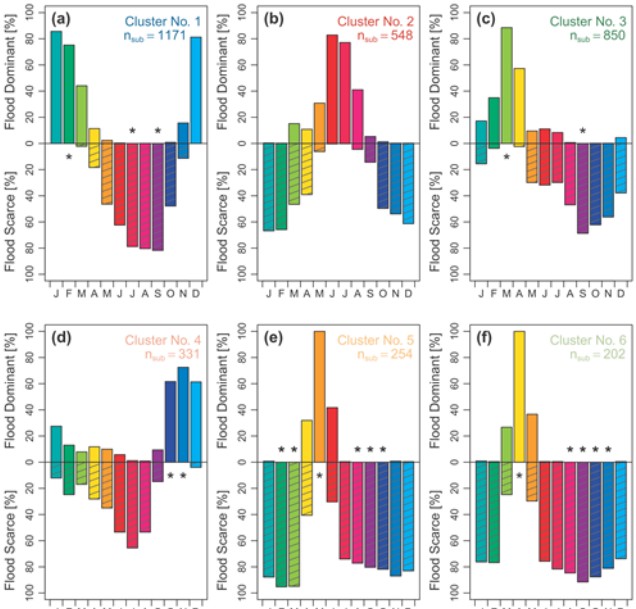

**Figure 13. Percent of stations with months that can be considered significantly (α=0.05) flood dominant (upward bars) or flood scarce (downward shaded bars) grouped by cluster (a-f). Months for which no station showed significance in the respective category are marked with a star. $n_{sub}$ indicates the number of stations included in the analysis (only stations with records > 30 yrs).**

Figure 14 shows, the maximum and minimum duration (in months) for flood dominant and scarce periods respectively. In each panel, the percentages are listed for each cluster separately. There is a very small number of stations (<1%) in Clusters 1, 2 and 4 for which no significant flood dominant season could be identified (i.e. 3, 2 and 2 stations, respectively) (Fig. 14a). This means that the floods are not uniformly distributed throughout the year (as stations for which uniformity could not be

15   rejected were already removed from this dataset in a previous step), but the number of floods per months does not cross the $L_{upper}^{n}$ threshold for the months to be classified as flood dominant. All clusters have a maximum of five consecutive flood dominant months, with exception of cluster 6, which has one station that has six consecutive months (Fig. 14a). Most of the stations have a maximum length of two consecutive months apart from Cluster 1, which has the highest number of three months (Fig. 14a). Clusters 5 and 6 have at least one flood scarce month, all other clusters have less than < 5% of their

20   stations without flood scarce month (Fig. 14c and d). Most of the stations have a minimum duration of 1 flood scarce month





(Fig. 14d). This is important, as the existence of flood scarce months is a necessary condition for the identification of bimodal flood seasonality distributions in the next step of the analysis. Most of the stations in Cluster 5 and 6 have the same maximum and minimum period duration of 2 months (Fig. 14a and b) and the same number of maximum and minimum length of flood scarce months between 9 and 10 months (Fig. 14c and d). This indicates that the seasonal flood seasonality distributions are

5    likely unimodal.

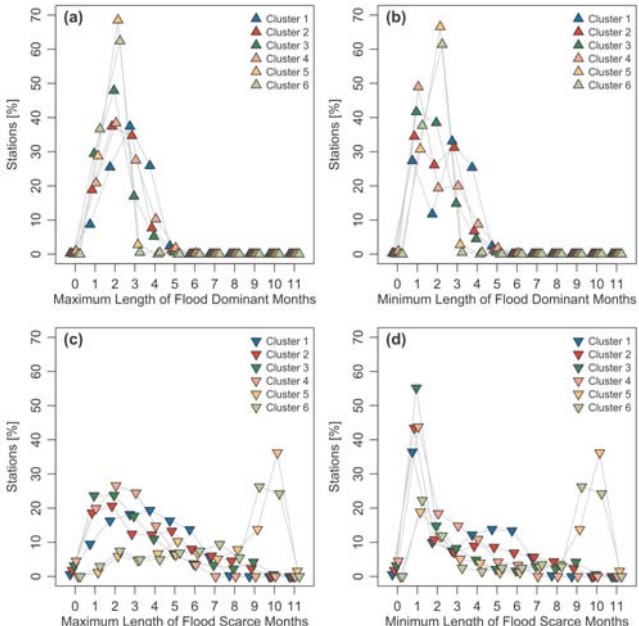

**Figure 14. Percent of stations within the same consecutive monthly flood classification, grouped per cluster. The x-axes show the maximum (left panels) and minimum number (right panels) of consecutive months classified as flood dominant (upper row) and flood scarce (lower row).**

Based on the alternating occurrence of flood dominant and flood scare periods or the uninterrupted occurrence of flood dominant periods, the flood seasonality distributions of 79 stations are classified as bimodal and 2490 stations as unimodal (Table 3). Cluster 4 has the highest number of stations (29) and the highest percentage of stations (~9 %) in a cluster with bimodal distributions. Cluster 4 has also the highest percentage of stations without a clearly defined flood seasonality

15    distribution and the lowest number of unimodal stations (~56 %). This indicates that Cluster 4 is the cluster with the most diverse flood seasonality distributions, which is consistent with its low average silhouette values detected before. In Cluster 2 and Cluster 3, 20 and 29 stations are classified as bimodal, the other clusters contain few bimodal stations.



**Table 3.** Six clusters and their characteristic flood seasonality distributions, based on the subset of station with records > 30 yrs.

| No. of k cluster | Location | No. Stations > 30 years | Bimodal | | Unimodal | | Undefined | | Primary Flood Season | Secondary Flood Season* |
|---|---|---|---|---|---|---|---|---|---|---|
| | | | No. | [%] | No. | [%] | No. | [%] | | |
| 1 | Western, Central and Southern Europe | 1171 | 1 | 0.09 | 861 | 73.53 | 309 | 26.39 | Dec to Mar | - |
| 2 | Mountainous regions | 548 | 20 | 3.65 | 393 | 71.72 | 135 | 24.64 | May to Aug | Mar |
| 3 | Central and Eastern Europe | 850 | 24 | 2.82 | 604 | 71.06 | 222 | 26.12 | Feb to Apr | Jun & July |
| 4 | Western British-Irish Isles, Western Coast of Norway and Northern Mediterranean | 331 | 29 | 8.76 | 184 | 55.59 | 118 | 35.65 | Oct to Jan | May |
| 5 | Northern Europe | 254 | 4 | 1.57 | 249 | 98.03 | 1 | 0.39 | May & Jun | - |
| 6 | North-Eastern Europe | 202 | 1 | 0.50 | 199 | 98.51 | 2 | 0.99 | Apr | - |
| All | Europe | 3356 | 79 | 2.35 | 2490 | 74.2 | 778 | 23.45 | - | - |

*Only for bimodal flood seasonality distributions in clusters with at least 5 stations.

In Figure 15, the monthly bimodal, unclassified and unimodal flood seasonality distributions are shown for each cluster separately. Primary and secondary flood seasons are identified, for each cluster separately, if the median of the monthly flood percentage is > 8.33% (1/12). Primary flood seasons are based on all stations with at least 30 years and secondary flood seasons are identified from the bimodal flood seasonality distributions, from cluster with at least 5 bimodal stations (excluding the months that are already included in the primary flood season) (Table 3).

Cluster 2, Clusters 3 and Cluster 4 show a monthly seasonality distribution with a distinct secondary flood season (Fig. 15.b.1 to 15.d.1). In the mountainous regions (Cluster 2), the secondary peak in the bimodal flood seasonality distribution in March precedes the main flood season in summer. In Central and Eastern Europe (Cluster 3) the main flood season in February to April is followed by secondary flood in June and July and in Cluster 4 the primary flood season in October to January is followed by an additional flood dominant month in May. 98% of the stations in Cluster 5 and Cluster 6 in Northern and North-Eastern Europe are classified as unimodal. In Cluster 1, 74% of the stations have unimodal flood distributions and one stations is classified as bimodal. The other stations have an unclassified seasonality distribution, which is mainly due to the absence of an additional month classified as either flood dominant or flood scare.





**Figure 15. Boxplot of the percentage of floods per month for each station of all stations with at least 30 years of data, grouped by their cluster (a-f) and by their annual flood seasonality distribution being bimodal, undefined or unimodal (x.1 - x.3 respectively).**

5   **The top and bottom of the boxes show the 75th and 25th percentiles (i.e. the upper and lower quartiles) respectively, whiskers extend to 1.5 × interquartile range beyond the box, the black band indicates the median, and outliers are shown as points. Panels containing less than 5 stations show points instead of boxplots.**





Figure 16 shows the spatial pattern of the stations with a bimodal or unimodal seasonality distribution. In Figure 16a one can see that the R-value of the bimodal stations is low (mean R-value of all bimodal stations is 0.35). However, bimodality does not necessarily imply a low concentration around the mean, if, for instance, the two flood seasons are separated by only one flood scarce month. The station with the highest R-value of all bimodal stations (R=0.73) is located in Finland (Cluster 5), for which the secondary flood season occurs just 2 months before the primary flood season. Even though bimodality in the flood seasonality distribution is only detected in a small number of stations, the locations of these stations are not randomly distributed across Europe, but rather located in close spatial proximity.

In Northern and Eastern Europe (Cluster 5 and Cluster 6), the duration of the period of flood dominant months (unimodal seasonality distribution) is the shortest (Fig 16b) and lasts, on average 1.73 and 1.65 months, respectively. The stations with a unimodal flood seasonality distribution in Cluster 1 have, on average, the longest period duration (~3.2 months). In Cluster 2, Cluster 3 and Cluster 4 the periods of flood dominant months last, on average, 2.57, 2.16 and 2.65 months respectively.

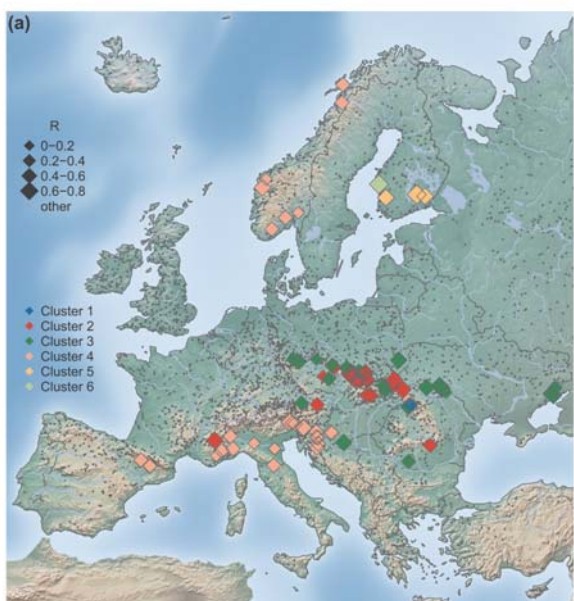

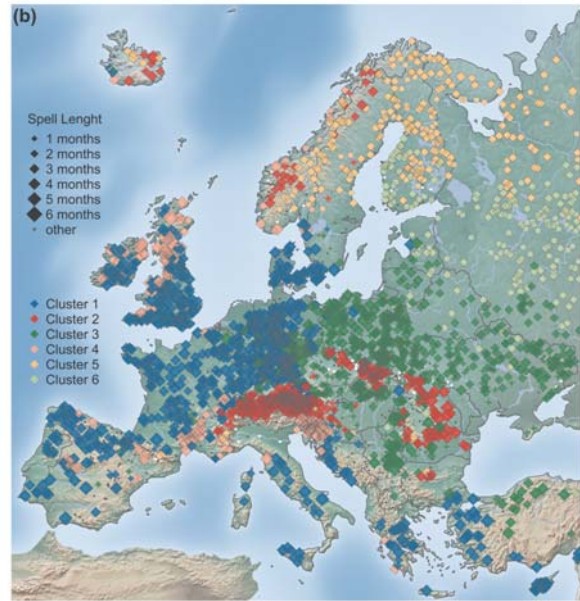

**Figure 16. Spatial distribution of the stations with bimodal flood seasonality distributions with the point size scaled by concentration R (a), and stations with unimodal distributions with the point size scaled by the length of the flood dominant period (b). Stations with bimodal distributions are marked by small white points shown in panel b).**



## 5 Discussion and Conclusions

This study provides a detailed analysis of the seasonality characteristics of annual maximum floods in Europe. While many previous studies at the national or regional scale, this paper aims aim at identifying large-scale geographical regions with similar temporal flood characteristics.

Previous studies in the US (e.g. Lecce (2000)) have suggested that catchment area has a strong effect on the flood seasonality (higher frequency in summer and autumn due to short duration summer storms of limited areal coverage). In Europe, this does not seem to be the case. Smaller catchments show little difference in their flood seasonality when compared to larger catchments. Here a difference is only apparent in summer floods, which is apparent for the larger catchments located in Eastern Europe. From the results presented one can conclude that in Europe the station elevation (i.e. the catchment outlet

elevation), or the catchment area explains the timing of the flood occurrence to a lesser degree than geographical location. It would have been interesting to investigate if the catchment mean or maximum elevation would correlate better with the observed flood seasonality patterns and clusters. However, this information was only available for few of the stations and could be analysed if such information becomes available.

The clustering was performed on the monthly frequency of the AMF occurrence. Even without considering the geographical

location in the clustering, larger-scale spatial coherent clusters emerged that had distinct characteristics with regard to their flood seasonality distributions. The spatial patterns detected are similar to those of smaller scale studies (some of which use different methods). For example, the clusters detected in this study had similar spatial boundaries as the 3 clusters identified by Beurton and Thieken (2009) in Germany. Cunderlik et al. (2004) detected 3 regions with different flood seasonality in Great Britain, whereas this study identified 2 clusters. Their region with a high number of floods in November (flood type 1)

corresponds approximately to cluster 4 identified in this study and their flood type 3 (floods occurring on average in January) corresponds to Cluster 1. They considered flood type 2 a transitional type between type 1 and type 3, which is included in cluster 1 of in study. However, differences between local scale and continental scale analyses are expected, as the differences in the monthly flood frequencies that appear to be important at a smaller scale may be of less importance at a larger scale where larger differences in the monthly flood seasonality distributions exist due to the existence of a larger variety of flood

generation processes.

Based on the mean flood seasonality, the temporal concentration of the AMF around the mean timing and the geographical location on the map, one can hypothesise the causes behind the observed patterns. For example, Cluster 5 and Cluster 6 in Eastern and North-Eastern Europe are likely to be predominately driven by snow melt processes (Blöschl et al., 2017), which result in a high temporal concentration within a month due to the relatively fast melting of the snow once the temperatures

rise in the spring. Compared to Cluster 5 and Cluster 6, Cluster 3 is located further to the South and the West. These locations are aligned with earlier snowmelt and a stronger maritime influence, respectively, which cause the floods to occur more frequently early in the year and exhibit a stronger influence of winter precipitation. For Cluster 2, which is primarily located in and around mountainous regions, one can infer that the AMF are caused by both snow and glacier melt in the summer, and





by heavy summer precipitation, occurring in two or three summer months. In the coastal areas with strong maritime influence located in Cluster 1, station elevation has little influence on the temporal occurrence of the annual maximum floods, as snow accumulation and melt is scarce. Therefore, this cluster can be considered to be mainly driven by extreme precipitation in late winter and early spring (Blöschl et al., 2017).

Cluster 4 is the most geographically dispersed cluster of all. The stations of Cluster 4 are located at the western coast of the British-Irish Isles, the western coast of Norway and the northern coasts of the Mediterranean. The temporal distribution of the AMF within the year shows a bimodal distribution for some stations of this cluster. The floods can be considered to be predominately driven by late autumn and early winter precipitation, but also contain some floods caused by spring and early summer precipitation. In this study, only the annual maximum floods were available, but if more than one flood per year

would be analysed (e.g. using partial duration series) the number of stations with bimodal distributions would probably be higher, as secondary flood maxima of a year would be included.

This study identifies spatially distinct regions with characteristic temporal patterns of temporal flood occurrence. A transition in the pattern of mean seasonality is apparent, from winter floods in Western Europe to late spring and early summer floods in Eastern Europe, onto which (depending on the region) late spring to summer floods are superimposed. The temporal

concentration of floods around the mean date of flood occurrence is highest in North and North-Eastern Europe and on the western lower latitude coasts. This is also apparent in the low temporal spread of floods (on average less than two months) and the high occurrence of stations with a unimodal flood season in the clusters located in these regions. The occurrence of a bimodal flood seasonality distribution over the year is only detected in a small number of stations, therefore, bimodality in the temporal distribution of AMF is therefore considered a local phenomenon of spatially distinct locations in Europe.

Nevertheless, in these regions the existence of a distinct secondary flood season is of practical importance for reservoir and flood risk management.

Overall, one can conclude that, for most of stations, the geographical location (including elevation) and hence regional climate is the most important factor influencing the timing of annual maximum floods in Europe. Therefore, the study can be considered a contribution towards advancing the understanding of geographical and climate sensitivity of annual maximum

floods and their temporal characteristics across Europe. Due to the strong spatial consistency of the clusters obtained, the results will also be important for an improved understanding of flood generation mechanisms at the European scale and the insights on the flood seasonality characteristics gained in this study provide a benchmark for the assessment of European-wide hydrological model output.





*Data Availability.* The data analysed in this study is based on the database from the works of Hall et al. 2015 and Blöschl et al. 2017 with additional some updates.

*Competing Interests.* The authors declare that they have no conflict of interest.

5   *Acknowledgements.* The research was supported by the ERC Advanced Grant "FloodChange" Project No. 291152. All calculations and Figures were produced using R (R-Core-Team, 2016). The following packages used in the study are acknowledged: 'classInt', 'cluster', 'fpc', 'maptools', 'plot3D', 'plotrix', 'plyr', 'raster', 'RColorBrewer', 'rgdal', and 'rworldmap'. The authors would like to acknowledge relevant discussions with Rui Perdigão and thank all data providers and fellow researchers involved in the data preparation.

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
