# Peer review of "Spatial Patterns and Characteristics of Flood Seasonality in Europe"

_Hydrology and Earth System Sciences, 2017_

## Referee Comment (RC1) · K. Breinl (Referee) · 12 Dec 2017

This is a very well-written and interesting paper that covers a lot of different aspects of flood seasonality across the European continent. The beauty of the study lies in its large spatial scale and the large number of different aspects that the authors consider. Another aspect of novelty relates to the distribution of flood seasonality, as up until today the mean date has been in focus, which can be misleading in the context of flood generating mechanisms.

As – in my eyes – there are no serious issues that can be addressed, I would like to highlight some more general suggestions for improvement. What I miss a bit is a more pronounced link of the results presented to the meteorology. I think it would

make the paper even better if the authors tried to link their observations to dominant precipitation patterns and/or weather situations across Europe. The authors highlight the strong influence of the flood timing to geographical location and, indeed, some aspects have been addressed in the discussion and conclusions (Page 25 Line 26 to Page 26 Line 11). However, some more information on the dominating weather/climate processes would contribute even more to the goal of "advancing the understanding of geographical and climate sensitivity of annual maximum floods (. . .) across Europe" (Page 26 Line 24-25). At this stage, I find the related section a tad too short, especially when considering the efforts behind all the different analyses presented.

Having mentioned the link to meteorology above, facing a warming climate, it may be interesting to shortly address how the flood seasonality may change in the next decades across the clusters and what this could mean from a disaster risk perspective, i.e. "better flood estimation and forecasts" as the authors state. There are some obvious changes with increasing temperature such as earlier spring floods in mountainous catchments, but maybe there are some aspects the authors can address for Europe of which the reader may not be instantaneously aware of. This could be a short paragraph.

What I would also recommend is for example to add some aspects on the applicability of the results in practise. The authors mention better flood estimation and forecasts but I think it may be good to add a few sentences that are a bit more specific.

Further comments:

- Page 6 Line 9: Can you be a bit more quantitative and mention how small the differences of the R-Value are considering shorter and longer observation records?

- As the number of gauges considered vary in the different analyses, it may be helpful to add the number of gauges considered in a short sentence to the figure caption ("n of m stations were considered" for instance)

- Page 11, L 10-11: Can you try to explain why you are facing the phenomenon of an unclear distinction in these transitional areas? Explanation on the phenomenon is given for mountainous areas thereafter but it would be important to find some explanation also for the other less distinct areas

- Page 11 Line 18-20. For the sites with low R-values but no uniformity (assuming skewed or bimodal distributions), have you looked into the actual distributions for confirmation?

Minor comments:

- Page 1 Line 10 I suggest "at continental scale"

- On page 2, figure 1 is mentioned in line 5 but at this point it is not mentioned yet that 4105 of the 5565 gauges are analysed. This is a bit misleading as the figure says 4105 gauges. I would recommend reordering the text so that the reader already knows about the 4105 gauges (i.e. cleaned database) when the text refers to figure 1

- Page 2 Line 24 I suggest "at continental scale"

- Page 2 Line 30 I suggest for better readability "(. . .) characteristics, and are followed (. . .)"

- Page 3 Line 18/19 I would reorder and first mention figure 2a and then figure 3

- Page 14 Line 4 "Figure 8 depicts"

- Page 20, Line 11: "Figure 14 shows the (. . .)"

- Figure 9 is not referenced in the text

Comments on figures:

The figures are of very high quality in general, but I hope that the resolution of the figures will be higher in the final paper as it is not ideal (too low) in the discussion paper.

- In general, many figures refer to the colours for the mean timing of floods as presented in Figure 4a (e.g. Figure 6). If possible, you may consider adding the colour circle to each figure for example as a bar legend below. I think it would improve the readability

- Figure 1. You may consider using darker cycles (black) or dots as the spatial distribution in some regions (e.g. Scandinavia) is a bit difficult to detect

- Figure 8b. No idea if this is possible, but can the European boundaries added to the bottom of the plot, which would be really nice?

- Figure 9. You may consider making all text black as it is difficult to read with brighter colours, especially when printing it (same for figure 11, 12, 13, 15)

- Figure 9, 11, 12: maybe you can explain "n" in the caption

Sincerely yours,

Korbinian Breinl (Uppsala University, Sweden)
* * *

---

## Referee Comment (RC2) · Anonymous Referee #2 · 22 Dec 2017

Dear authors,

This is an extremely interesting study that I would like to see published in Hydrology and Earth System Sciences journal. This manuscript presents an exhaustive and comprehensive spatial and temporal analysis of flood seasonality at European scale through the identification of region with similar characteristics. I really enjoyed reading the paper, which deals with the important and timely issue of notable interest and modernity, especially for the HESS readership. The paper accurately presents the methods and results. I have just a few minor comments/suggestions for the authors to consider.

1. In this manuscript, the authors always refer to flood. As mentioned in section 2 (study area and data), only annual maximum discharge or water level are used to run the statistical analyses. However, it is not always the case that annual maximum values

convert to flood. Could the authors clarify this issue and explain how it may affect their findings?

2. As stated by reviewer 1, I suggest the authors to more details regarding the correlation between spatiotemporal flood pattern and meteorology.

3. The analyses on the mean flood seasonality and temporal flood concentration look very similar to the one recently proposed by the same authors in Blöschl et al. (2017). The authors have to clearly state the differences between the analyses in these two papers. If there are no differences, I recommend them to shorten or remove the results description to give more space to the ones on the characterization of flood spatial patterns.

4. Because of the amount of information and figure (16) it is sometimes difficult to follow the description of the results and grasp the main take-home message. I suggest the authors to summarize and select the main findings and key figures.

5. Dots in Figures 1, 2 and 3 are quite difficult to read because of the topographic map used as background. In addition, no legend is provided. I recommend to make cleared figures or remove them if not necessary.

Reference:

Blöschl, G. et al. (2017) Changing climate shifts timing of European floods, Science.

---

## Author Comment (AC1) · 8 Feb 2018

This is a very well-written and interesting paper that covers a lot of different aspects of flood seasonality across the European continent. The beauty of the study lies in its large spatial scale and the large number of different aspects that the authors consider. Another aspect of novelty relates to the distribution of flood seasonality, as up until today the mean date has been in focus, which can be misleading in the context of flood generating mechanisms.

As – in my eyes – there are no serious issues that can be addressed, I would like to highlight some more general suggestions for improvement. What I miss a bit is

a more pronounced link of the results presented to the meteorology. I think it would make the paper even better if the authors tried to link their observations to dominant precipitation patterns and/or weather situations across Europe. The authors highlight the strong influence of the flood timing to geographical location and, indeed, some aspects have been addressed in the discussion and conclusions (Page 25 Line 26 to Page 26 Line 11). However, some more information on the dominating weather/climate processes would contribute even more to the goal of "advancing the understanding of geographical and climate sensitivity of annual maximum floods (...) across Europe" (Page 26 Line 24-25). At this stage, I find the related section a tad too short, especially when considering the efforts behind all the different analyses presented.

Response: Linking the observed spatial and temporal patterns of floods in Europe to the dominating weather/climate processes is indeed an important research topic. However, given the variety of processes associated to floods across Europe we belief that such a detailed analysis would be beyond the scope the current study and rather merits a separate study that builds on the results from the current manuscript

Having mentioned the link to meteorology above, facing a warming climate, it may be interesting to shortly address how the flood seasonality may change in the next decades across the clusters and what this could mean from a disaster risk perspective, i.e. "better flood estimation and forecasts" as the authors state. There are some obvious changes with increasing temperature such as earlier spring floods in mountainous catchments, but maybe there are some aspects the authors can address for Europe of which the reader may not be instantaneously aware of. This could be a short paragraph.

Response: It is indeed of importance to investigate how the flood timing might change in future, however we belief that this would go beyond the scope of the current manuscript. The nature of future changes in the annual maximum flood timing across Europe is still understudied (albeit the existence of a few regional studies) and therefore we believe that the analysis of future changes deserves a detailed study on its

own. This could for example be done in combination with the detailed analysis of the current dominating weather/climate processes and how these are projected to change in future.

What I would also recommend is for example to add some aspects on the applicability of the results in practise. The authors mention better flood estimation and forecasts but I think it may be good to add a few sentences that are a bit more specific. Response: We will elaborate on this in the revised version.

Further comments: - Page 6 Line 9: Can you be a bit more quantitative and mention how small the differences of the R-Value are considering shorter and longer observation records?

Response: Further elaboration will be added.

- As the number of gauges considered vary in the different analyses, it may be helpful to add the number of gauges considered in a short sentence to the figure caption ("n of m stations were considered" for instance)

Response: Will be added.

- Page 11, L 10-11: Can you try to explain why you are facing the phenomenon of an unclear distinction in these transitional areas? Explanation on the phenomenon is given for mountainous areas thereafter but it would be important to find some explanation also for the other less distinct areas.

Response: The explanation is already given in L 11-12. "the AMF of these stations tend to occur in March and April around the cut off date separating the winter- versus summer half-years."

- Page 11 Line 18-20. For the sites with low R-values but no uniformity (assuming skewed or bimodal distributions), have you looked into the actual distributions for confirmation?
Response: Yes, we performed a visual inspection and distribution was indeed not uniform. In L 20 the word likely will be removed to increase clarity.

Minor comments: - Page 1 Line 10 I suggest "at continental scale"

Response: Article in the sentence will be removed as suggested.

- On page 2, figure 1 is mentioned in line 5 but at this point it is not mentioned yet that 4105 of the 5565 gauges are analysed. This is a bit misleading as the figure says 4105 gauges. I would recommend reordering the text so that the reader already knows about the 4105 gauges (i.e. cleaned database) when the text refers to figure 1

Response: The first reference to Fig 1 will be removed and the one to Fig 2b will be moved further down to avoid misinterpretation. Additionally, the Figure caption already states the number of stations that is shown in Fig 1.

- Page 2 Line 24 I suggest "at continental scale"

Response: Article will be removed as suggested.

- Page 2 Line 30 suggest for better readability "(...) characteristics, and are followed (...)"

Response: Placement of commas will be adjusted as suggested.

- Page 3 Line 18/19 I would reorder and first mention figure 2a and then figure 3

Response: Order will be swapped as suggested.

- Page 14 Line 4 "Figure 8 depicts"

Response: Parenthesis will be removed as suggested.

- Page 20, Line 11: "Figure 14 shows the (...)"

Response: Comma will be removed as suggested.

- Figure 9 is not referenced in the text

Response: The Fig. 9 is already mentioned on page 14, line 5.

Comments on figures:

-The figures are of very high quality in general, but I hope that the resolution of the figures will be higher in the final paper as it is not ideal (too low) in the discussion paper.

Response: Figures will be of high quality in the final version. Low resolution is due to the import into Word.

- In general, many figures refer to the colours for the mean timing of floods as presented in Figure 4a (e.g. Figure 6). If possible, you may consider adding the colour circle to each figure for example as a bar legend below. I think it would improve the readability

Response: Legend for coloured points will be added in Fig 6.

- Figure 1. You may consider using darker cycles (black) or dots as the spatial distribution in some regions (e.g. Scandinavia) is a bit difficult to detect.

Response: We prefer not to use filled circles (i.e. dots) as this covers the underlying topography and does not allow to depicting station density. Instead we will make the colour of the circles darker to increase visibility.

- Figure 8b. No idea if this is possible, but can the European boundaries added to the bottom of the plot, which would be really nice?

Response: Country borders will be added to the 3D plot.

- Figure 9. You may consider making all text black as it is difficult to read with brighter colours, especially when printing it (same for figure 11, 12, 13, 15)

Response: For better readability the coloured text will be printed in bold letters in all Figures. This allows text to be read without difficulties in the final publication with high quality resolution.
- Figure 9, 11, 12: maybe you can explain "n" in the caption

Response: Explanation will be added.

---

## Author Comment (AC2) · 8 Feb 2018

Dear authors, This is an extremely interesting study that I would like to see published in Hydrology and Earth System Sciences journal. This manuscript presents an exhaustive and comprehensive spatial and temporal analysis of flood seasonality at European scale through the identification of region with similar characteristics. I really enjoyed reading the paper, which deals with the important and timely issue of notable interest and modernity, especially for the HESS readership. The paper accurately presents the methods and results. I have just a few minor comments/suggestions for the authors to consider.

1. In this manuscript, the authors always refer to flood. As mentioned in section 2

(study area and data), only annual maximum discharge or water level are used to run the statistical analyses. However, it is not always the case that annual maximum values convert to flood. Could the authors clarify this issue and explain how it may affect their findings?

Response: In hydrological practice, floods are commonly defined as the largest observed flow in a given year, and are widely used (flood frequency analysis in the Flood Estimation Handbook). However, by definition, this does not necessary mean that the annual maximum flood always overtops the river banks. A sentence with this detail will be added to the flood definition to avoid possible misinterpretations of the results by readers outside the hydrological community.

2. As stated by reviewer 1, I suggest the authors to more details regarding the correlation between spatiotemporal flood pattern and meteorology.

Response: Linking the observed spatial and temporal patterns of floods in Europe to the meteorology is indeed an important research topic. However, given the variety of meteorological conditions associated to flood generation in Europe we believe that such a detailed analysis is beyond the scope the current study and rather merits a detailed separate study that builds on the results from the current manuscript.

3. The analyses on the mean flood seasonality and temporal flood concentration look very similar to the one recently proposed by the same authors in Blöschl et al. (2017). The authors have to clearly state the differences between the analyses in these two papers. If there are no differences, I recommend them to shorten or remove the results description to give more space to the ones on the characterization of flood spatial patterns.

Response: The paper by Blöschl et al. (2017) focuses on the changes/trends in the timing of floods. A map of the mean flood seasonality was used in that paper to put the observed changes in timing in context. In the current manuscript the mean seasonality is also calculated but the analysis goes more into detail (e.g. separate detailed analysis

of the mean timing and the temporal concentration of the floods around the mean). This detailed analysis cannot be removed as it provides the background information necessary to contextualise the other results presented in the paper.

4. Because of the amount of information and figure (16) it is sometimes difficult to follow the description of the results and grasp the main take-home message. I suggest the authors to summarize and select the main findings and key figures.

Response: The legend in Figure will be amended to detail the information better that Figure 16 is displaying. Additionally, in the final version of the manuscript, we will add a section detailing the main finding.

5. Dots in Figures 1, 2 and 3 are quite difficult to read because of the topographic map used as background. In addition, no legend is provided. I recommend to make cleared figures or remove them if not necessary.

Response: The presence of the maps in Figure 1-3 with the schematic topography in the background is important information, which is necessary for interpretation and spatial and topographic contextualisation of the study. To increase the contrast with the background map we will add borders to the non-overlapping points. Additionally, in final print version the figures will also appear clearer due to the increased resolution.

---

## Author Response (AR2)

**Editor Decision:** by Giuliano Di Baldassarre

Comments to the Author:

The manuscript was significantly improved by the authors, but a few comments were not sufficiently addressed. The paper can be published after a minor revision, which should consider the following three points.

In our previous response to the referee comments, we had explained why certain comments/requests made by the referees are out of scope of this study.
The points made by the referees are all important research areas; however, to answer these questions extensive additional research at a European scale is needed to obtain a complete and non-speculative answer. We now discuss these aspects and the areas for future research in detail in the discussion section.

1. Both referees comment about the fact that the paper would benefit from a better link to dominant precipitation patterns and/or weather situations across Europe. I agree with them. More information on the dominating weather processes would contribute even more to the goal of "advancing the understanding of geographical and climate sensitivity of annual maximum floods".

We agree that such links would be important to obtain, however such links between the obtained spatial and temporal patterns of flood seasonality cannot readily be made.
To elaborate this, we provide a detailed discussion why a link/attribution of the obtained flood seasonality patter to the climate/weather and precipitation patterns is beyond the scope of this study and why it merits a detailed follow up research. Such research should focus exclusively on the links between climate, flood generation processes, and the spatial and temporal patters of flood seasonality.

2. The authors should give more credit to other scholars and previous work in the field. This paper would benefit from more references to previous studies, as not many studies are cited here.
All sections that were added to discuss the content of the manuscript, contain additional references to previous studies.

This can be achieved by
discussing the role of precipitations patterns (point 1 above),
See answer to point 1 above

discussing potential future possibilities (point 3 below),
See answer to point 3 below

or adding a discussion about the actual implication of flood seasonality (why is it relevant?) by linking this study more with current advances in the study of dynamic flood risk (socio-hydrology, practical applications, etc.).
In the discussion section, we included an elaboration on the importance of this European wide-floods seasonality study, and the practical and scientific importance.

All these additional aspects would enrich this paper, and make it stand out more from previous recent publications.

3. I also agree with another point raised by Rev#1. One single paragraph discussing potential changes in the future would make this paper much more interesting. This does not imply additional work, but a scientific discussion about the potential role of increasing temperature. If the authors think that nothing can be said about future changes, it would be very interesting to read exactly why.
We detail the potential changes of flood seasonality in future, together with a scientific discussion on the current limitations in projecting flood seasonality changes in the discussion section.